# Assessing and Improving Provider Knowledge for a Cardiothoracic Intensive Care Unit Electronic Dashboard Initiative

**DOI:** 10.3390/healthcare11081136

**Published:** 2023-04-15

**Authors:** April Garlejo, Jacob Bonner, Ashley Paddock, John Park, Nolan Lyda, Ahmed Zaky, Susan McMullan

**Affiliations:** 1Post-BSN to DNP Nurse Anesthesia, School of Nursing, University of Alabama at Birmingham, 1701 University Blvd., Birmingham, AL 35294, USA; 2Department of Cardiac Anesthesiology and Critical Care Medicine, University of Alabama at Birmingham Hospital, 1802 6th Avenue S., Birmingham, AL 35233, USA

**Keywords:** electronic dashboard, educational training, ICU quality indicators

## Abstract

Background: Electronic dashboards measure intensive care unit (ICU) performance by tracking quality indicators, especially pinpointing sub-standard metrics. This helps ICUs scrutinize and change current practices in an effort to improve failing metrics. However, its technological value is lost if end users are unaware of its importance. This results in decreased staff participation, leading to unsuccessful initiation of the dashboard. Therefore, the purpose of this project was to improve cardiothoracic ICU providers’ understanding of electronic dashboards by providing an educational training bundle in preparation for an electronic dashboard initiation. Methods: A Likert survey assessing providers’ knowledge, attitudes, skills, and application of electronic dashboards was conducted. Subsequently, an educational training bundle, consisting of a digital flier and laminated pamphlets, was made available to providers for four months. After bundle review, providers were assessed using the same pre-bundle Likert survey. Results: A comparison of summated scores from pre-bundle (mean = 38.75) and post-bundle surveys (mean = 46.13) yielded an increased summated score overall (mean = 7.38, *p* ≤ 0.001). Conclusion: An educational bundle improved providers’ understanding and increased their likelihood of using electronic dashboards upon its initiation. Further studies are needed to continue increasing staff participation such as providing specific education to navigate the interface for data retrieval and interpretation.

## 1. Introduction

Intensive care units (ICU) are complex, data-rich environments that require extensive documentation of quality indicators that are used as metrics to reflect patient status and also serve to measure unit performance. Quality metrics include, but are not limited to, occurrences of re-intubations, amount of time on the ventilator, blood transfusions, length of stay, catheter-associated urinary tract infections, and central line bloodstream infections. Monitoring these occurrences is crucial, as these metrics directly affect patient outcomes and hospital expenses [1]. The inability to identify and monitor declining quality indicators, due to lack of integration and accessibility, hinders early interventions and worsens patient outcomes [2,3]. The Joint Commission reported 120 sentinel events, with 23% being clinical content design related issues and 23% being communication issues from health information technology [4]. Therefore, ICUs are likely to implement a system that gathers quality metric data over a certain period of time and present these data trends in a meaningful display for analysis. Such a monitoring system often requires manual collection of extensive data from the cumbersome, inaccessible formatting of electronic medical records [2,3]. This burden of manually finding, inputting, and organizing data in various records has fallen onto providers who already have a significant workload [5]. Fortunately, the advent of electronic dashboards is a solution that the current literature recognizes. An electronic dashboard is a computerized system that automatically collects quality metric data and transparently disseminates the data in a meaningful, online display for providers to easily analyze unit performance [6]. 

Electronic dashboards are used in many ICUs throughout the United States, yet not all ICUs have successfully implemented the novel technology, including a cardiothoracic intensive care unit (CICU) in a Southeastern academic medical center located in a major metropolitan area. This particular CICU had plans to initiate a new electronic dashboard and approached the doctoral team to help the integration of this technology to its staff. The doctoral team was debriefed by CICU leadership regarding several factors that led to failed initiation and sustainability of a previous electronic dashboard on this particular CICU (A. Zaky, MD, oral communication, February 2021). However, for academic time-sensitive purposes, the doctoral team chose to focus on the issue pertaining to the CICU staff’s lack of knowledge regarding the novel technology. The CICU leadership team recognized and agreed with this decision to focus on the educational issue.

An extensive literature search regarding electronic dashboards and system changes was conducted using the resource library search engines PubMed and CINAHL. These search engines provided the current literature that emphasized that an electronic dashboard is a streamlined method used to transparently disseminate electronic medical records and is currently a valuable tool used in many hospitals to improve quality metrics [6]. However, changes to any practice in a hospital will meet resistance [7,8]. To mitigate resistance, the literature supports completing an assessment of knowledge and likelihood of change in the population of interest prior to large-scale changes. Identification of areas of improvement were evaluated to tailor the provided education. The improvement areas were then reassessed to determine whether or not the actual implementation and sustainability of the change will be successful. Through the review of the literature, our team recognizes that evidence-based interventions, such as electronic dashboards, may be ineffective if staff are not appropriately educated on its significance in relation to how it affects the staff’s care practices and their patients’ outcomes [9].

To parallel these literature suggestions, the doctoral team will use the focus, analyze, develop, and execute (FADE) model to guide the project. The FADE model is a widely known and accepted framework of quality improvement (QI) in healthcare to test changes for QI studies [8,10]. It was the framework of choice for this project because the model outlined systemic elements to elicit responsiveness to the intervention being implemented. As shown in Figure 1, the project was conducted chronologically, in the order of focus, analyze, develop, and execute. The focus of this QI project was to assess improvement of the CICU providers’ knowledge of electronic dashboards through surveys. The analysis of baseline knowledge was conducted using a pre-intervention survey that highlighted providers’ knowledge gaps regarding electronic dashboards. The doctoral team then developed a plan to create and implement an educational training bundle, based on the knowledge gaps identified in the pre-intervention survey. This project was executed between May 2022 and August 2022. A subsequent evaluation, comparing results from the pre-intervention and post-intervention surveys, determined whether improvements in the knowledge gaps occurred after providing the educational training bundle. 

This QI project aimed to improve CICU providers’ understanding of electronic dashboards by providing an educational training bundle to prepare for the CICU’s initiation of an electronic dashboard. Creating a dashboard requires agreement and cooperation between multiple stakeholders. This includes approval from hospital administrators, unit managers, and the quality improvement committee. In addition, from a logistical standpoint, it requires close communication, time, and professionals competent in information technology to create a dashboard tailored to a CICU’s needs. Therefore, our intent is that the outcome of this project will provide sufficient evidence and data to our key stakeholders to further pursue the implementation of an electronic dashboard.

## 2. Materials and Methods

The doctoral team utilized a digital survey website called Qualtrics to input the survey questions. Qualtrics is an online platform that utilizes tailored surveys, projects, and data to generate statistical information that project teams and organizations can then disseminate their findings. This platform is protected by user login information that is personal to the individual user, and project data is only shared to those individually selected to access the data or project determined by the original creator. Qualtrics is a global online company with twenty offices that span five continents. Qualtrics is headquartered in Seattle, Washington, United States, and was founded in the year 2002. The team monitored the progress, accuracy, and completeness of the surveys throughout the duration of the project. By doing so, the project cost was able to be minimized through the free access of Qualtrics through the team’s academic university services, and without a third party monitoring and collecting the survey responses.

This project designed pre-bundle and post-bundle surveys, using a single group to determine the effectiveness of an educational training bundle. The CICU nurse educator disseminated numerous one-page informational sheets within the CICU providers’ work area. These sheets contained three quick reference (QR) codes: a pre-bundle survey, an educational training bundle, and post-bundle survey. All three QR codes were provided at the beginning of project implementation to encourage thorough completion of this project, and they remained available from 1 May 2022 to 31 August 2022. The inclusion criteria for participation in the project included holding an unencumbered license, actively working full-time (0.9 full-time employment) or part-time (0.6 full-time employment) in a CICU patient care setting, age > 19 years old, English language speaking and writing ability, all genders, and all races. Licensure held within the unit consisted of registered nursing (RN) licenses, with degrees ranging from a Bachelor of Science in Nursing (BSN) to an Associate Degree in Nursing (ADN). Advanced practice provider licenses consisted of a Certified Registered Nurse Practitioner (CRNP), Physician’s Assistant (PA), or a Certified Registered Nurse Anesthetist (CRNA). The advanced practice providers held Masters degrees or Doctorates in their respective fields. Lastly, physicians with either a Medical Doctorate (MD) or Doctor of Osteopathic medicine (DO) held board certifications in cardiac anesthesiology, intensive care, or cardiology.

The educational training bundle consisted of two forms of distribution. Digital and physical forms of the training bundle were used to improve knowledge gaps observed in the pre-bundle survey. These two platforms of delivery were chosen because the digital platform allowed easy access for education and survey delivery to the participants, and the physical pamphlet provided educational sustainment after the project. The digital education material was accessible using a QR code located on the informational sheet disseminated by the CICU nurse educator. This consisted of standard dashboard components, the importance of monitoring quality metrics, and how dashboards help improve current care practices and optimize patient outcomes. The physical, laminated educational pamphlets listed a summary of key points that paralleled those in the digital education material, serving as the continued educational support for CICU providers to review beyond the completion of this project. These provided education materials will be used by the CICU team to establish the foundation of the implemented electronic dashboard, thus reinforcing the obtained knowledge by the healthcare team.

The current literature highlights an assessment of the population’s knowledge and its likelihood to change prior to any large-scale modifications, as it will alleviate the resistance when change is implemented [7,8]. The Planned Behavior Theory (PBT) is a cognitive theory that predicts one’s decision to engage in a particular behavior based on his or her intention to engage in that behavior [11,12]. Therefore, the stronger the intention to engage in the behavior, the more likely it will be performed [11,12]. We believed that this theory would help measure the degree of CICU providers’ behaviors of acceptance and adoption of innovative change in the medical realm, especially after providing education on why and how the electronic dashboard affects care practices and patient outcomes. Therefore, the PBT was used as a guide for the construction of the pre-bundle and post-bundle surveys, as shown in Figure 1. 

The pre-bundle and post-bundle surveys were identical. The first four questions were a demographic section, which included the participants’ initials, age, type of licensure and position, and full-time equivalence. The participants’ self-generated unique identifiers allowed the doctoral team to match data results from the pre-bundle and post-bundle surveys. Following the demographic section, the next 12 questions consisted of a PBT-influenced behavioral section assessing CICU providers’ attitudes, knowledge, skills, and application of electronic dashboards. The surveys utilized a 5-point Likert scale, in which participants selected answers that reflected varying degrees of agreement: strongly agree, somewhat agree, neither agree nor disagree, somewhat disagree, and strongly disagree. A numerical value was assigned to each answer (5 = strongly agree, 4 = somewhat agree, 3 = neutral, 2 = somewhat disagree, and 1 = strongly disagree) for numerically comparing the pre-bundle and post-bundle survey results to reveal the degree of impact that the educational training bundle had on the CICU providers. 

### Analysis

Survey results were analyzed and assigned numerical values for each varying degree of agreement on the Likert scale. The counts are used in Figure 2. Numerical values were analyzed using the International Business Machines Corporation Statistical Package for the Social Sciences, version 28. Figure 3 illustrates all participants’ pre-bundle and post-bundle survey summated scores, and Figure 4 displays the input scores in a paired *t*-test, yielding a statistically significant *p*-value of <0.001. The doctoral team recognized the small sample size of this project, therefore a non-parametric Wilcoxon test was used, as the literature recommends, to further fortify and affirm the results. The non-parametric Wilcoxon test also reflected a statistically significant *p*-value of <0.001 [13]. The surveys’ 12-question behavioral section was grouped into four categories: attitude, knowledge, skills, and application. Within each category, the summated scores were inputted into a paired *t*-test, from which the mean and standard deviations were derived. 

## 3. Results

Sixteen CICU staff members were surveyed voluntarily, and all participants met the inclusion criteria. The average age of participants was 28 years old. The majority of participants were registered nurses with a Bachelor of Science in Nursing or an Associate Degree in Nursing (68.8%), whereas other participants included certified registered nurse anesthetists with a Doctorate in Nursing Practice (18.8%) and nurse practitioners with a Master of Science in Nursing (12.4%). A total of 75% of participants worked in the CICU, whereas 25% had previously worked in the CICU. The participants’ questionnaire paired *t*-test yielded summated pre-bundle survey (mean = 38.75 and standard deviation = 8.714) and post-bundle survey (mean = 46.13 and standard deviation = 9.507) scores with a *p*-value of <0.001. The doctoral team omitted one survey question from the analysis because the question was structured as a double negative, which confused participants and potentially skewed the data. 

Summated pre-bundle and post-bundle survey scores were obtained for each question, and each question was grouped according to the four aforementioned categories. Questions related to attitude resulted in a pre-bundle survey mean of 57.67 and standard deviation of 3.215, and a post-bundle survey mean of 71.67 and standard deviation of 0.577. Questions related to knowledge resulted in a pre-bundle survey mean of 62.5 and standard deviation of 4.95, and a post-bundle survey mean of 71.5 and standard deviation of 0.707. Questions related to skills resulted in a pre-bundle survey mean of 54.33 and standard deviation of 2.082, and a post-bundle survey mean of 62.33 and standard deviation of 2.309. Questions related to the application resulted in a pre-bundle survey mean of 53 and standard deviation of 8.544, and a post-bundle survey mean of 65.33 and standard deviation of 5.033.

## 4. Discussion

### 4.1. Interpretation

The results inputted into a paired *t*-test reflected an increased mean score of 7.38 points when the pre-bundle and post-bundle surveys were compared. The results yielded a statistically significant *p*-value of <0.001. Furthermore, a non-parametric Wilcoxon test was used due to the small sample size, and it also showed a statistically significant *p*-value of <0.001. According to these results, our educational training bundle positively correlates with the improvement of healthcare provider knowledge of electronic dashboards of those who fully participated in this project. The doctoral team further evaluated the mean score and the standard deviation of the questions to assess their attitudes, knowledge, skills, and application. The mean scores for attitude, knowledge, skills, and application improved by 14, 9, 8, and 12.33 points, respectively. Therefore, the CICU providers’ intent towards electronic dashboard use increased significantly when they were educated regarding the technology’s benefits as a means of improved care delivery and tracking quality indicators [3,10]. This improvement in the staff’s baseline knowledge increased the likelihood of their acceptance and willingness to adopt towards unit change. These key findings were consistent with the current literature: to decrease resistance to change, education of the population of interest must occur prior to implementation to ensure successful integration and sustainability of that change [7,8]. 

### 4.2. Implication

The doctoral team’s main objective in this project was to improve the CICU providers’ understanding of electronic dashboards in preparation for the technology’s actual implementation on the unit. The team has done so by directly assessing and measuring the staff’s understanding of electronic dashboards, with special attention given to their attitudes, knowledge, skills, and application of this novel technology. The findings suggest that staff education of electronic dashboards before its actual implementation is key to the technology’s successful integration on the unit. This is because the adequate education, which addressed staff’s areas of weakness resulting from the initial assessment, provided CICU providers a greater understanding of the rationales and urgency for the novel electronic dashboard. 

One practical contribution of the project’s findings is that it further strengthens the current literature regarding education, prior to actual initiation of a change, which leads to greater success of its acceptance, adoption, and sustainability. Adequate education is key to helping the affected individuals better understand why and how the change leads to improvements to their systems of process at the individual level, group level, and organizational level. Therefore, leaders must focus on education as the most effective catalyst for change. For example, ICUs contain multiple complex care practices that are constantly changing, especially during and after the COVID-19 pandemic. In order for new care practices to take effect immediately, the ICU leadership team must focus on developing effective educational resources for ICU staff to understand the urgency for the change taking place, particularly addressing why and how the change will affect an individual’s care routine. If the new care practices are being exercised individually, the unit’s performance may improve its associated quality indicators over time and help improve the hospital’s overall cost-effective delivery of care and patient satisfaction ratings.

Additionally, ICU non-leadership individuals, such as bedside nurses, respiratory therapists, and physical therapists, could apply the project’s findings to improve their delivery of quality care. By focusing on educating the who, what, where, when, why, and how of the new adjustments being made prior to its actual initiation, the chances of its adoption will improve. For example, the bedside nurse may act as a ‘leader’ to open heart surgical patients. Prior to the bedside nurse helping ambulate the patient around the unit post-operatively, the nurse informs and educates the patient on the importance of early ambulation to reduce post-operative atelectasis and swelling. By providing the patient with rationales as to why he or she should walk the next day after surgery, the bedside nurse helps motivate the patient to follow the post-operative care plan to mitigate post-operative complications. Since the patient has a greater understanding of the bedside nurse’s care plan, the patient is thus more likely to accept early ambulation and adopt this change during their hospital stay. 

Aside from the project’s contributions, this project has a strength that allows readers the opportunity to easily replicate this project on their units. The use of surveys as the method to collect data provided the team with the ability to gather information from a large group at a low cost. The pre- and post-assessment surveys on Qualtrics easily provided a representation of the CICU providers’ understanding of electronic dashboards. The ease of creating the surveys on Qualtrics, distributing the surveys by QR codes, allowing participants continuous access to complete the surveys, and securely collecting the data in real time was time-effective and cost-effective for the team.

Additionally, this project had various limitations that hindered different aspects of the project. First, the lack of CICU provider project participation led to the small sample size (*n* = 16) at the time of the project initiation. This project was initiated during the COVID-19 pandemic, which may have contributed to work fatigue and high staff turnover within the CICU where the project was implemented. The unprecedented stress on experienced providers at the time of project implementation had numerous unforeseen consequences. The increasing responsibilities placed on the providers, coupled with a high rate of staff burnout, led to voluntary participation in a new project being classified as a low priority. A larger sample size across multiple ICUs would have been beneficial and is recommended by this doctoral team in future studies. Second, the doctoral team was unable to physically visit the CICU to publicize the project’s initiation because of the hospital’s COVID-19 policies, thereby relying solely on the CICU nurse educator and this project’s clinical mentor to encourage provider participation. Perhaps emailing all CICU providers a short, informational Zoom video recording of the doctoral team announcing the QI project would have been beneficial and is thus recommended by this project team in the future. Third, the educational training bundle was written in English. There is a possibility that some CICU providers may be proficient in speaking English; however, they may not be proficient in reading and comprehending university-level English. This barrier can be further investigated to evaluate the reading level of the educational training bundle and adjust it to the level appropriate for the majority of the staff that also adequately conveys the information provided. 

To continue provider engagement with the electronic dashboard, further studies are needed to provide specific education regarding the navigation of the digital interface for data retrieval and interpretation. If providers do not understand how to find desirable data within the computerized system, or find the process of data retrieval cumbersome, then they may be less likely to use the electronic dashboard despite understanding its technological benefits. Therefore, continuous refinements must be made to the electronic dashboard upon implementation. These refinements should be completed in a dynamic fashion in conjunction with the healthcare providers to address the majority of the target population’s issue with the novel technology. The goal is to better serve our healthcare team in a more user-friendly manner for prompt utilization. 

## 5. Conclusions

The implementation of novel health technologies has significantly improved patient care and patient outcomes. Transparency and education are key to acceptance of new health technologies. Presentation of evidence regarding how the innovative care practice of electronic dashboards as a means of tracking quality indicators positively impacts patient care delivery encourages staff adoption. This QI project assessed and improved providers’ knowledge of electronic dashboards in preparation for eventual implementation within the CICU. Providing the educational training bundle for initial education, with continual access to the bundle following project completion, allowed CICU providers to become familiar with the novel technology and served as a reference for sustainability. Sustainability for this project includes staff accessibility to educational material at any given time with the use of the provided QR codes, as the actual implementation of the electronic dashboard approaches in the near future. This will support and guide direction for electronic dashboards to track quality indicators in order to improve unit performance and patient care delivered. 

## Figures and Tables

**Figure 1 healthcare-11-01136-f001:**
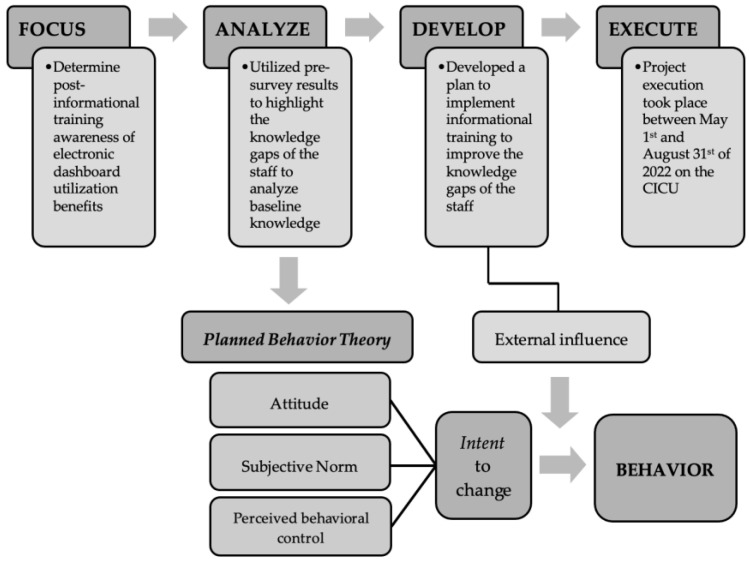
Adapted from FADE Model and Theory of Planned Behavior [9,10,11,12]. Reprinted/adapted from Ref. [https://josieking.org/patientsafety/module_a/methods/fade.html]. 2002–2021, Duke University.

**Figure 2 healthcare-11-01136-f002:**
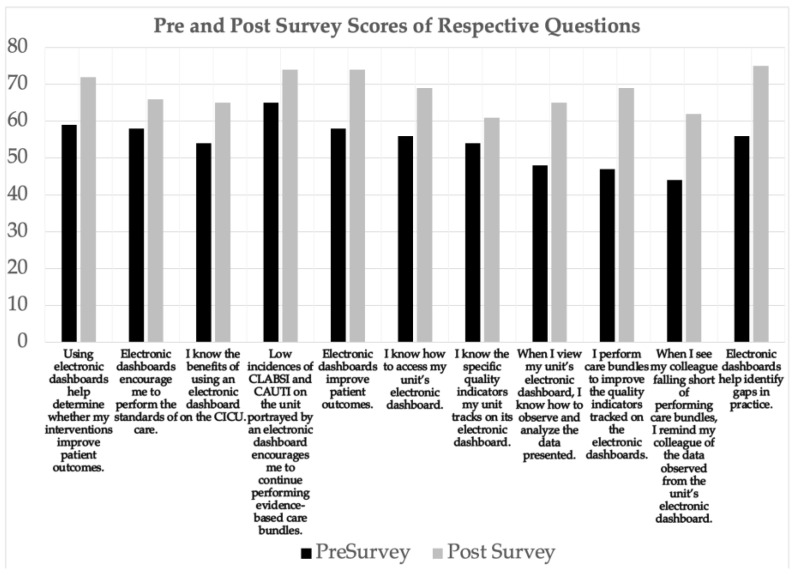
Measurement of respective question scores.

**Figure 3 healthcare-11-01136-f003:**
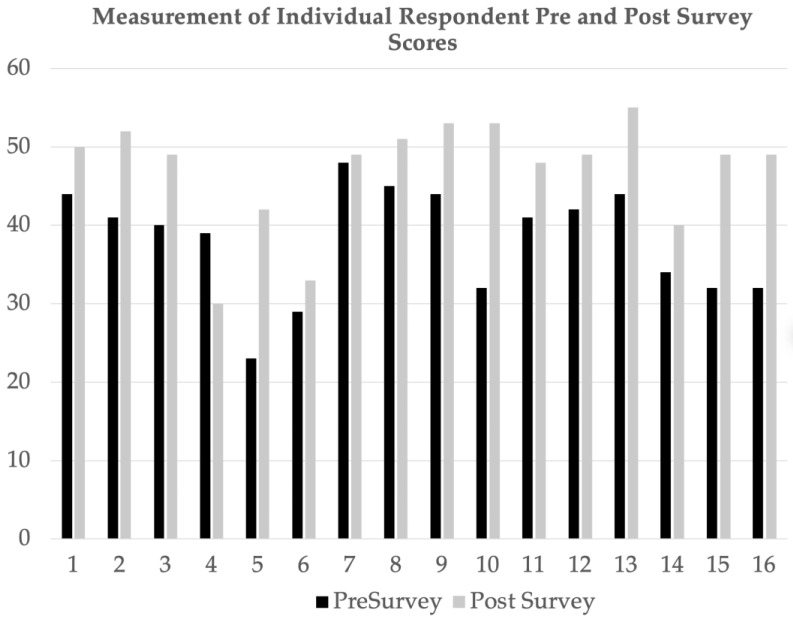
Measurement of individual participant scores.

**Figure 4 healthcare-11-01136-f004:**
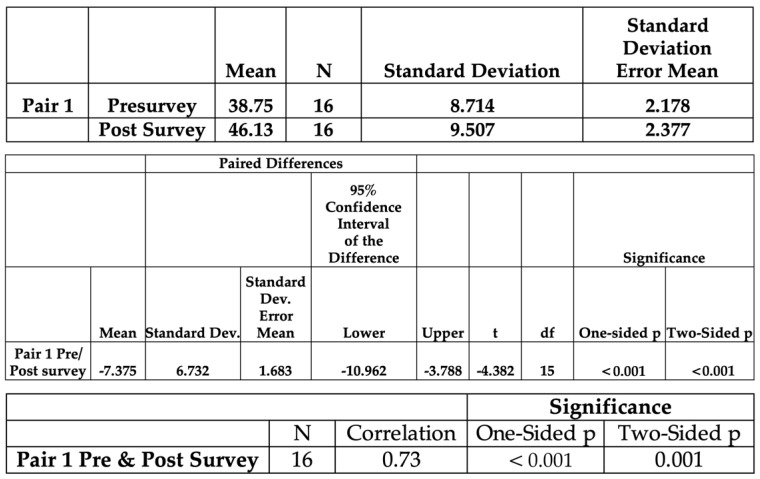
Pre-survey and post-survey paired *t*-test.

## Data Availability

The data presented in this project are available upon request from the corresponding author. The data is not publicly available due to privacy policies the team has set forth during planning.

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
