# Peer review of "Assessing and Improving Provider Knowledge for a Cardiothoracic Intensive Care Unit Electronic Dashboard Initiative"

_healthcare, 2023, doi:10.3390/healthcare11081136_

Round 1

Reviewer 1 Report

The article by April garlejo et al presents a new approach to improve users' knowledge of electronic dashboards,The reason is that the electronic dashboard measures ICU performance by tracking quality indicators, particularly pinpointing those that are below the norm.

This facilitates the ICU to scrutinize and change current practices in an effort to improve indicators of failure. They adopted the methodology of a Likert survey assessing provider knowledge, attitudes, skills, and application of electronic dashboards. The vendor was then provided with an educational training package consisting of a digital flying machine and lamination booklet for a period of four months. After bundle review, providers assessed using the same pre bundle Likert survey.

The purpose of this paper is to hope to adopt this approach to let users know the importance of electronic dashboards, improve their technical value and increase staff engagement leading to a successful dashboard startup.

This paper is well written and well-organized, and has great inspiration on how to improve our understanding of electronic dashboards. Therefore, I recommend that the following questions be addressed prior to publication of this article:

(1) References titled 1, 2, 5 as well as 8 are incorrectly formatted and the literature names should all follow the title.

(2) Line caption lettering in Figure 2 is too small and too pale for clarity to appear and should be sized appropriately to font color.

(3) Fourth, more than one full stop before figure 577 in line 162, and the same holds true for figure 707 in line 164.

(4) It is important to note that your manuscript needs to be carefully edited by someone proficient in English language technical editing, paying particular attention to English grammar, spelling, and sentence structure so that readers can clearly understand the objectives and results of the study.

Reviewer 2 Report

Review: Assessing and Improving Provider Knowledge for a Cardiothoracic Intensive Care Unit Electronic Dashboard Initiative

Abstract

1: Abbreviations should be introduced

Material and Methods

2: Educational training bundle: Please describe the package in more detail with all the contents. Why were two ways of distribution used?

3: Planned Behavior Theory (PBT): This paragraph is unclear. Why this theory? What makes it fit to the use cases examined here?

Analysis

4: Using mean scores for Likert Scales is inappropriate. The mapping of integer values to levels of agreement is arbitrary and does not suffice the use of a (paired) t-test. Please see: Kampen J, Swyngedouw M: The Ordinal Controversy Revisited. Quality and Quantity 2000, 34(1):87-102.

Results

5: Inclusion criteria are now mentioned but were never introduced.

6: Figure 3 is hardly readable.

7: Figure 4 represents an unformatted screenshot of SPSS output.

8: The results do not relate to the aim of this study.

Discussion

9: What does: “The results inputted into a paired t-test reflected an increased mean score of 7.38 points and a p-value of <0.001.” mean? The inappropriately used paired t-test revealed a difference in means of 7.35.

10: Paragraph Limitations appears twice.

Reviewer 3 Report

This manuscript describes a QI project which successfully implements an electronic dashboard and per pre-and post-implementation surveys, is able to raise awareness of the progress of QI efforts. 

Regarding the introduction, the first two paragraphs nicely set up the project to be discussed. Consider condensing the third paragraph and the described literature search which seems like it would be a given step for the project. 

For the "analysis" section of methods, would recommend taking out the question that introduces this section. The numerical assignments for the likert scale seem redundant to the previous section. 

The results clearly show the presented change from pre to post surveys. No edits suggested here. 

I would definitely like to see the discussion expanded as the implications of the electronic dashboard and its effects can be further explored. 
